# Impact of Cropland Evolution on Soil Wind Erosion in Inner Mongolia of China

Wenfeng Chi [1,2,†], Yuanyuan Zhao [3,*], Wenhui Kuang [4], Tao Pan [5,6], Tu Ba [4], Jinshen Zhao [3], Liang Jin [1,2] and Sisi Wang [7,†]

1 College of Resources and Environmental Economics, Inner Mongolia University of Finance and Economics, Hohhot 010070, China; cwf@imufe.edu.cn (W.C.); nmjinliang@163.com (L.J.)

2 Resource Utilization and Environmental Protection Coordinated Development Academician Expert Workstation in the North of China, Inner Mongolia University of Finance and Economics, Hohhot 010070, China

3 Key Laboratory of Soil and Water Conservation and Desertification Combating, Ministry of Education, School of Soil and Water Conservation, Beijing Forestry University, Beijing 100083, China; zhaojinshen_bjfu@163.com

4 Key Laboratory of Land Surface Pattern and Simulation, Institute of Geographic Sciences and Natural Resources Research, Chinese Academy of Sciences, Beijing 100101, China; kuangwh1@igsnrr.ac.cn (W.K.); laibatunacun@163.com (T.B.)

5 College of Geography and Tourism, Qufu Normal University, Rizhao 276826, China; pantao@qfnu.edu.cn

6 Land Research Center, Qufu Normal University, Rizhao 276826, China

7 National Remote Sensing Center of China, Beijing 100036, China; wangsisi@nrscc.gov.cn

* Correspondence: yuanyuan0402@bjfu.edu.cn

† Both authors contributed equally to this work.

**Abstract:** Understanding soil erosion responses to cropland expansion/shrinking plays a crucial role in regional agriculture sustainability development in drylands. We selected Inner Mongolia, a typical water resource constraints region with acute cropland expansion, as the study area in China. Spatial cropland evolution and its impact on wind-driven soil erosion were investigated with the help of field sampling data, remotely sensed retrieved data, and the revised wind erosion model (RWEQ). Results showed that the cropland area of Inner Mongolia presented an increased growth trend, with a net increase area of 15,542.9 km² from 1990 to 2018. Cropland characteristics in Inner Mongolia presented continuous growth in its eastern region, basically constant growth in its central region, and declined in its western region. Most cropland declines occurred after 2000 when the Grain for Green project began, which means that acute cropland expansion happened from 1990 to 2000. The soil wind erosion modulus showed a net increase with cropland expansion. The reclamation of forests and grasslands contributed to an increase of 5.0 million tons of the soil wind erosion modulus, 80% of which was produced in the eastern part of the region. The conversion from croplands to grasslands/forests caused a decrease of approximately 2.7 million tons, 62% of which was in the east and 25% in the west of the region. Considering the constraints of water shortage and over-exploitation of groundwater, we provide a path based on a balance between "resource-production-ecosystem" to achieve ecologically sustainable agriculture development in the drylands of China.

**Keywords:** land-use/cover changes; soil wind erosion; cropland evolution; sustainable path; ecological barrier area of North China

## 1. Introduction

Croplands provide significant provisioning ecosystem services (e.g., food and fiber), regulating (e.g., flood control and carbon sequestration) and cultural services (e.g., scenic beauty and education) to human communities [1]. The protection and sustainable use of cropland resources is a significant guarantee for global food security, and it is the foundation for achieving the 2030 Sustainable Development Goals (SDGs) [2]. It is also

an important issue for global development in the Future Earth science plan (ICSU, 2020). Drylands extend over 45% of the earth's land surface and could increase by an additional 7% by 2100 due to global warming [3,4]. More than half of the earth's croplands are located in drylands [5]. Even though dryland farming has its high-risk and economical challenges, it is an essential part of the world's agriculture food system and will become even more significant because of population growth and the increasing demand for grain [6]. The deterioration of water resources and the trend of a warmer and drier climate pose a serious threat to agricultural production in drylands in the 21st century [7,8].

Cropland expansion and intensification involves a trade-off between food provision and environmental deterioration. Soil wind erosion is a serious problem that threatens land production and environmental quality in drylands [9]. Crops grown in drylands are generally without irrigation and tend to be water resource limited [10]. Soil erosion decreases soil fertility, which can negatively affect crop yields. It also sends soil-laden water downstream, which can create heavy layers of sediment that prevent streams and rivers from flowing smoothly, impacting surface water irrigation during paddy field growth season [11]. The long dormant season, along with strong winds in spring, makes exposed croplands more vulnerable to wind erosion, especially those without crop residues or wind breaks [12–15]. The wind erosion modulus of rain-fed cropland will continue to increase more than that of grassland under future climate change scenarios [16]. Land degradation caused by wind erosion reduces soil productivity and weakens ecosystem services such as soil conservation and crop production [17]. Many measures involving combating wind erosion and land degradation in croplands have been conducted across the world, such as the Grain for Green Project [18,19]. Many previous studies have tried to investigate the wind-blown erosion from cropland based on field observation, wind tunnel simulation or numerical modelling [13,20]. However, the rates and spatiotemporal characteristics of wind erosion reduction or increasements from cropland changes in drylands have been poorly studied.

China feeds approximately 22% of the world's population with only 7% of the world's croplands [8]. Inner Mongolia, which belongs to drylands, is one of China's 13 major grain-producing provinces [21]. Much of this province is an agro-pastoral ecotone, which is vulnerable to global change. Soil wind erosion is one of the main issues limiting the local sustainable development of agriculture, and it is intensified by agricultural activities such as over-cultivation and grassland reclamation [22]. Previous literature on cropland resources in Inner Mongolia are mainly focused on the land conversion process [23], the driving mechanism [24] and the path to agricultural sustainability [25,26]. The specific responses of soil erosion to cropland expansion or shrinkage are less reported.

Therefore, this study aimed to reveal the spatiotemporal patterns of cropland resources and the response of soil wind erosion to these changes in Inner Mongolia, China from 1990 to 2018. We examined the factors affecting the sustainable development of agricultural resources and explored ways to ensure the coordinated development of food security and environmental protection for the drylands of China.

## 2. Materials and Methods

### 2.1. Study Area

Inner Mongolia is located in North China (37°–53° N, 97°–126° E), and includes 12 prefecture-level administrative regions, with a total area of 1.18 million km$^2$ (Figure 1). It is mostly within arid/semi-arid climate zones. The annual mean temperature is 0–8 °C, and annual precipitation ranges from 50 to 500 mm, varying greatly among seasons and years and showing a declining gradient from east to west. The main landforms include plateaus, mountains, plains, valleys, and basins. There are five major deserts (Badaim Jaran, Bayan Ondor, Qubqi, Tengger, and Ulan Bul) and four sandy lands (Hulun Beir, Hunshandak, Mu Us, and Horqin) in Inner Mongolia [27].

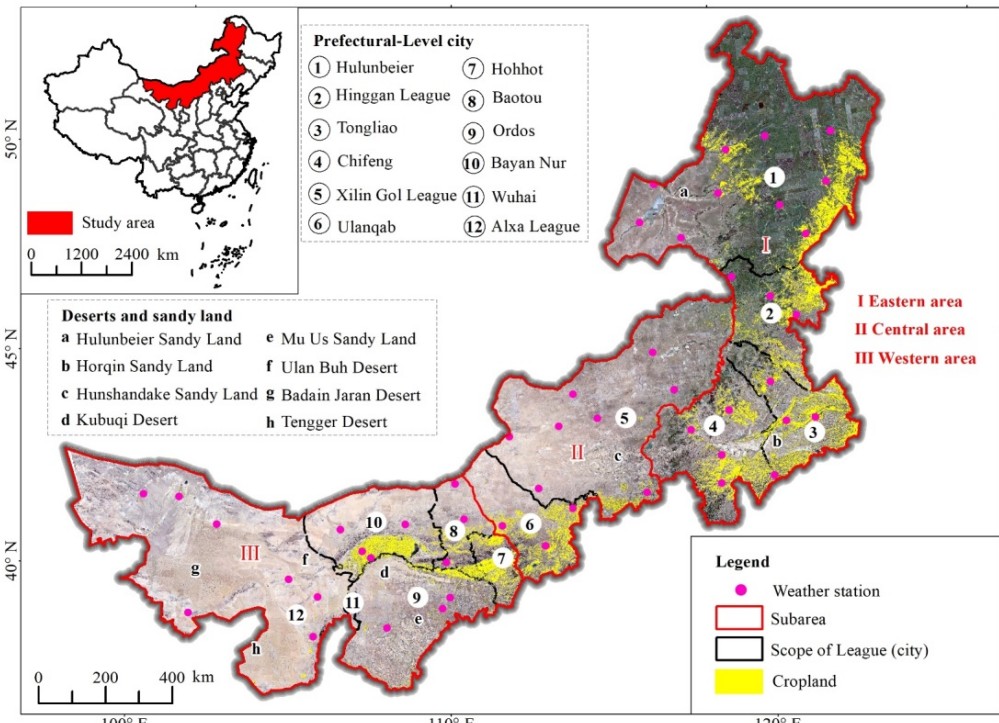

**Figure 1.** The location and cropland distribution of Inner Mongolia, China.

The study area was divided into three parts (i.e., the east, central, and west) based on the terrain, geomorphology, climatic conditions, and administrative units, in order to reveal the heterogeneity of cropland evolution and the corresponding wind erosion effects. The eastern part includes the cities of Hulun Buir, Xingan League, Tongliao and Chifeng, covering an area of approximately 0.49 million km$^2$. The central part includes the cities of Xilingol and Ulanqab, covering an area of approximately 0.26 million km$^2$. The western part includes the cities of Hohhot, Baotou, Wuhai, Ordos, Bayannur and Alathan, covering an area of approximately 0.45 million km$^2$. The study area has grassland ecosystems in the east and center, and the deserts and Gobi ecosystems in the west. The variation in natural conditions determines the different impacts of cropland evolution on soil erosion processes in Inner Mongolia.

### 2.2. Methods

The technical framework map of this study is provided in Figure 2. In this map, the information on field observations and measurements for validation process, measured parameters, and sampling locations selected is displayed. Specifically, field observations and measurements are displayed in Figure 2a. The stratified random sampling to obtain random verification points is used in Figure 2b. The localized parameters for this model are displayed in Figure 2c, and the coefficient of accuracy between 137Cs and RWEQ retrieval is displayed in Figure 2d.

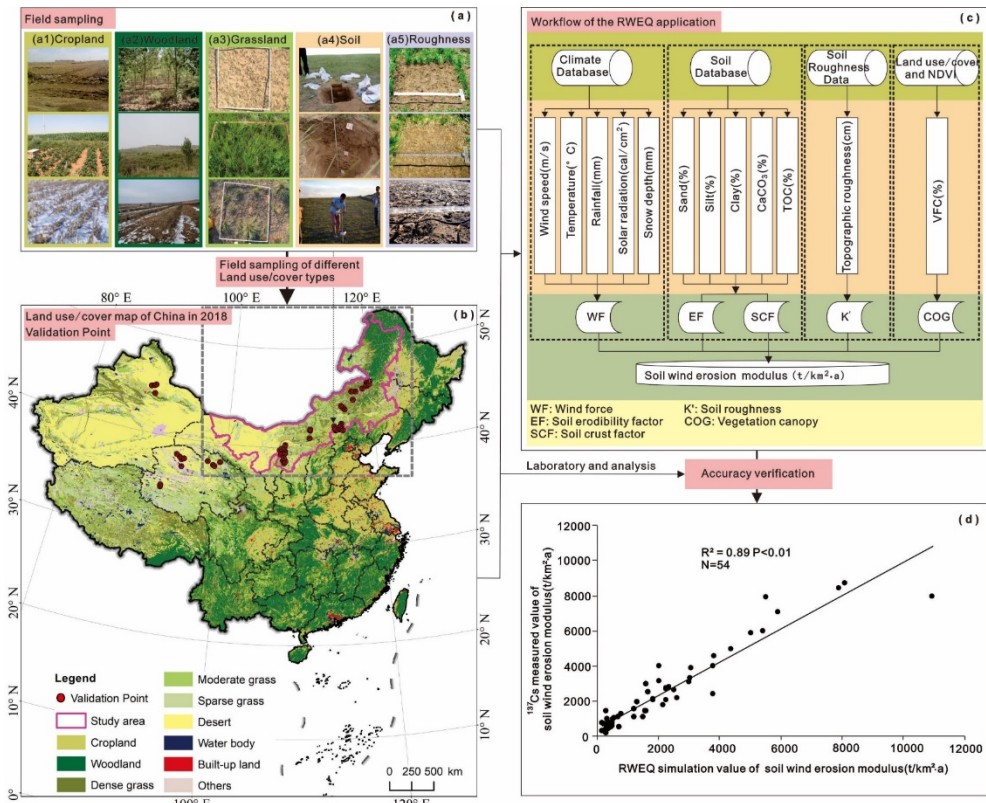

**Figure 2.** The retrieval and validation of soil wind erosion modulus data: (**a**) photographs of field sampling for accuracy verification, (**b**) spatial distribution of verification points, (**c**) workflow of the revised wind erosion equation (RWEQ) application, and (**d**) the coefficient of accuracy between 137Cs and RWEQ retrieval.

*2.3. Data*

The data used in this study included land-use/cover, the normalized difference vegetation index (NDVI), meteorological data, water resources and water utilization data, statistical data, and other auxiliary data. Land-use/cover data at a scale of 1:100,000 in 1990, 1995, 2000, 2005, 2010, and 2015 were obtained from the national China Land-Use/Cover Datasets (CLUDs) produced by the Chinese Academy of Sciences (http://www.resdc.cn/, accessed on 20 April 2021) [23,28]. There were six first-level classes, including cropland, forest, grassland, water, built-up land, and bare land. We updated the data using Landsat OLI and GF/2 (Gaofen-2 satellite) data in 2018. Classification accuracy was over 90% based on GF2 and GE (Google Earth) images with 1 m spatial resolution.

NDVI datasets with a spatial resolution of 8 km and a 15-day interval during the period 1990–2006 were downloaded from GIMMS3g, and those for 2000–2018 were obtained from MODIS (MOD 13A2) with a 1-km spatial resolution and a 16-day interval. The GIMMS and MODIS NDVI datasets were processed and composited using the maximum value composite method to obtain monthly and yearly NDVI data [29]. There was a statistically significant linear relationship in the NDVI between the GIMMS and MODIS NDVI time series during 2001–2006 at the regional scale. Time series of the NDVI were used for retrieving combined vegetation factors of wind erosion.

Meteorological observation data (i.e., the daily wind speed, precipitation, temperature and sunshine hours) covering 51 stations within/around Inner Mongolia were obtained from the China Meteorological Science Data Center (http://data.cma.cn/, accessed on 20 April 2021). Time series of meteorological data were interpolated using ANUSPLIN for the simulation of the climate factor of wind erosion. Statistical data, including crop planting data, were obtained from the Statistical Yearbooks of Inner Mongolia. Water resources and irrigation amounts were collected from the Water Resources Bulletin of Inner Mongolia.

Data on the soil type and texture, soil organic matter content, and soil organic matter were obtained from the Resources and Environment Data Cloud platform (http://www.resdc.cn/, accessed on 20 April 2021). The calcium carbonate content in soil was derived from the national Earth System Science Data Center (http://www.geodata.cn/index.html, accessed on 20 April 2021) China 1:400,000 soil calcium carbonate content distribution map. All of these were used for calculating the soil erodibility and soil crust factor of the cropland.

The land-use/cover changes were examined based on the vector data. The data for modeling the soil wind erosion were resampled to a spatial resolution of 1 km in order to facilitate data space matching and calculation. All operations were conducted on the ArcGIS platform.

### 2.4. The Detection of Cropland Changes

We detected cropland changes from three aspects: quantity, conversion type, and planting structure. The quantity was measured by the net annual change area and annual change rate (Equation (1)) [23]:

$$K_i = \Delta S_i / S_i / \text{t} \times 100\% \tag{1}$$

where $K_i$ is the annual change rate of cropland during period $i$, $\Delta S_i$ is the total net change area of cropland, $S_i$ is the cropland area at the starting year of the period $i$, and t is the time period. Then, the conversion from or to cropland were examined. We conducted the analysis at intervals of 5 years from 1990 and at a regional and sub-region scale.

### 2.5. Analysis of Soil Wind Erosion Dynamics with Cropland Evolution

The RWEQ model, which uses meteorological, soil, and vegetation factors as the main input data, was intended to be used for simulating the soil wind erosion of cropland [30]. It has been successfully used for large geographical scales and long-term wind erosion modeling in many regions through revision, and obtained good simulation results [31]. First, the annual soil wind erosion modulus based on the revised wind erosion equation (RWEQ) was calculated. The RWEQ model was widely used for soil wind erosion modeling in North China, and it considered climatic conditions, surface roughness, vegetation coverage, soil erodibility, and soil crust factors [32,33]. The basic equations are as follows [30]:

$$S_L = 2z/S^2 \cdot Q_{max} e^{-(z/s)^2} \tag{2}$$

$$S = 150.71 \cdot \left( \text{WF} \cdot \text{EF} \cdot \text{SCF} \cdot \text{K}' \cdot \text{COG} \right)^{-0.3711} \tag{3}$$

$$Q_{max} = 109.8 \left( \text{WF} \cdot \text{EF} \cdot \text{SCF} \cdot \text{K}' \cdot \text{COG} \right) \tag{4}$$

where $S_L$ represents the soil wind erosion modulus (kg·m$^{-2}$); $z$ is the maximum wind erosion occurrence distance downwind; $s$ is the key block length (m); $Q_{max}$ represents the maximum sandstorm transport capacity (kg·m$^{-1}$). WF stands for climate factor, and is calculated based on, for example, wind speed, precipitation, temperature, and sunshine hours. EF is the soil erodibility factor (dimensionless); SCF represents the soil crust factor (dimensionless), and they are modeled based on the soil texture and soil organic matter. K' represents the surface roughness factor (dimensionless), and COG represents combined vegetation factors (dimensionless), which was estimated based on remotely sensed NDVI and sampling data.

We localized related parameters and validated them by field investigation and literature reviews [22,31]. During field observations, the surface roughness on various land-covers was measured. Withered vegetation covers obtained from samplings were regressed with friction vegetation cover calculated from remotely sensed NDVI data in order to improve the accuracy of COG. Soil samples were collected at intervals of 0-2-4-6-8-12-18-24 cm for tracing 137Cs gain or loss. With the equations above, the wind erosion modulus from 1990 to 2018 was firstly computed at a time scale of 15-day, and then summed up to one-year based on ArcGIS platform support and Python programming language.

The validation showed that the simulated soil wind erosion modulus was significantly correlated with the results of 137Cs detection (P < 0.001) [33].

In order to eliminate interference factors on the soil erosion modulus, we calculated the annual soil erosion modulus from 1990 to 2018 using the average climate condition of this period. Then, the soil erosion modulus changes from cropland variation were evaluated using the following equation [34]:

$$E_i = \sum_{k=1}^{j} A_{i,\,k} \times (e_{i,\,m} - e_{i,\,n}) \tag{5}$$

where $E_i$ refers to the net soil wind erosion modulus change in region $i$; $j$ refers to the amount of change types related to cropland; $A_{i,k}$ refers to the area of land-use/cover change type in region $i$; $e_{i,m}$ and $e_{i,n}$ refer to the average value with land-use/cover $m$ (the end year, 2018) and $n$ (the start year, 1990) for the duration of the analysis in zone $i$ respectively.

## 3. Results

### 3.1. Spatio-Temporal Patterns of Cropland Evolution during 1990–2018

The total cropland area of Inner Mongolia was 118,467 km$^2$ in 2018, which accounted for 9.87% of the whole study area. A total of 62% of this (72,989 km$^2$) was located in the east, 17% (20,069 km$^2$) was located in the center, and the other 21% (25,409 km$^2$) was located in the west. Specifically, croplands from the east were mainly located in the eastern part of the Greater Khingan Mountains and the Horqin Sandy Land, while those from the center were mostly located in the southern part of Ulanqab. Croplands from the west were mainly located on the upper reach plains of Yellow River (Figure 1).

From 1990 to 2018, croplands in Inner Mongolia displayed an increasing growth trend, although there were spatial heterogeneities and temporal variations. Total net cropland increases were 15,542.9 km$^2$, with a newly reclaimed cropland area of 26,540.9 km$^2$ and a loss area of 10,998.1 km$^2$, respectively. The net annual change area declined from an increase of 1682.1 km$^2$/a to a slight increase (Figure 3, Table 1). Of the three sub-regions (i.e., the eastern, central and western parts of Inner Mongolia), croplands from the east showed a large and continuous growth trend, but the growth rate changed in different periods between 1990 and 2018. Newly reclaimed croplands were mainly located in the eastern part of the Greater Khingan Mountains, and the revegetated croplands were mainly located in Hulun Beir. Meanwhile, croplands from the center generally had a requisition-compensation balance and showed a continuous decline after 2000. In contrast, croplands from the west decreased annually from 1995. Locations with cropland loss showed an agglomerated pattern, but the reclaimed regions were scattered (Figure 3, Table 1).

**Table 1.** Cropland changes in different sub-regions during different periods.

| | East | | Center | | West | | Inner Mongolia | |
|---|---|---|---|---|---|---|---|---|
| Period | Annual Change Area (km²/a) | Annual Change Rate (%) | Annual Change Area (km²/a) | Annual Change Rate (%) | Annual Change Area (km²/a) | Annual Change Rate (%) | Annual Change Area (km²/a) | Annual Change Rate (%) |
| 1990–1995 | 1506.30 | 13.16 | 11.52 | 0.28 | 164.28 | 3.24 | 1682.10 | 8.17 |
| 1995–2000 | 1215.51 | 9.38 | 205.07 | 4.84 | −178.04 | −3.51 | 1242.53 | 5.58 |
| 2000–2005 | 231.37 | 1.63 | −69.30 | −1.71 | −4.49 | −0.09 | 157.58 | 0.67 |
| 2005–2010 | 71.72 | 0.50 | −9.89 | −0.24 | 8.53 | 0.16 | 70.37 | 0.30 |
| 2010–2015 | 123.99 | 0.86 | −117.30 | −2.89 | −60.25 | −1.16 | −53.55 | −0.23 |
| 2015–2018 | −1.50 | −0.01 | −2.66 | −0.04 | 20.10 | 0.24 | 15.94 | 0.04 |
| 1990–2018 | 562.14 | 27.49 | 3.31 | 0.45 | −10.34 | −1.14 | 555.10 | 15.10 |

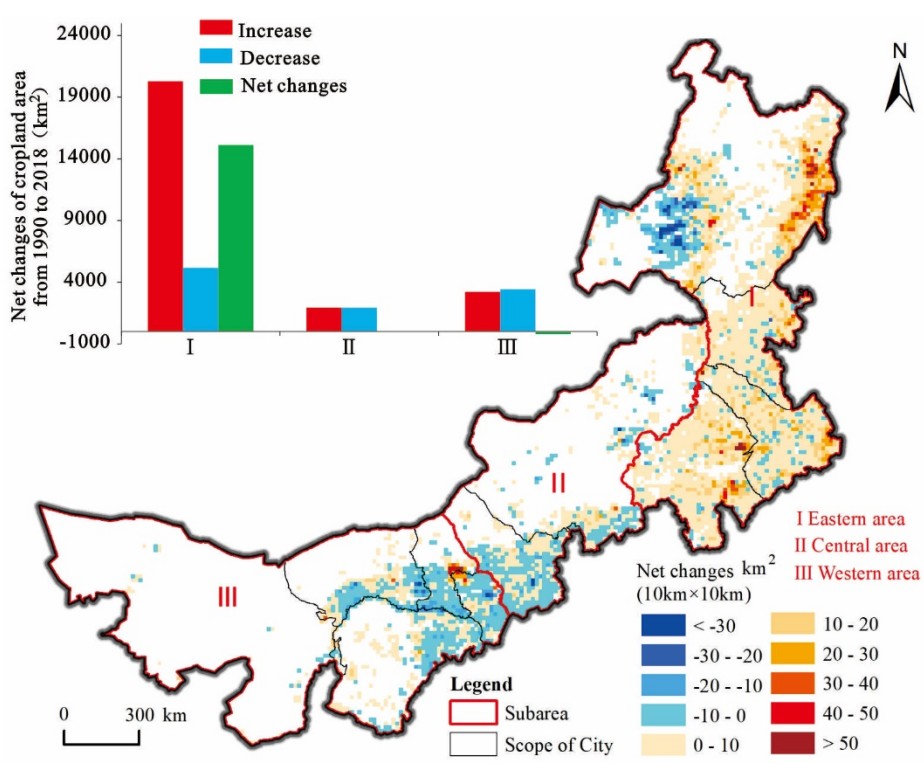

**Figure 3.** Spatial pattern of cropland evolution in Inner Mongolia of China from 1990 to 2018.

### 3.2. The Tracked Cropland Conversions in the Three Sub-Regions from 1990 to 2018

The newly expanded croplands from 1990 to 2018 were converted from previous forests, grasslands, and bare lands, with the greatest conversion area from grasslands. First, in the eastern part of Inner Mongolia, the croplands converted from grasslands were 8598 km², 3437 km² and 1651 km² in the yearly ranges of 1990–1995, 1995–2000, and 2000–2005, respectively, which accounted for 81%, 51%, and 70% of the total conversion cropland area during the corresponding periods, respectively (Figure 4). These reclamation areas were mainly distributed in the cities of Hulun Beir, Tongliao, and Chifeng. On the contrary, decreases in croplands in this sub-region were mainly through revegetation to grassland and forests in Hulun Beir. Second, in the central sub-region, an area of 1525 km² was converted from grasslands to croplands during 1990–2018, and most of these areas came into existence from 1995 to 2005. Further, about 1300 km² of other croplands have converted to grasslands since 1995. Third, in the west, a total area of 2645 km² of grasslands were reclaimed from 1990 to 2005, but 2024 km² of croplands returned to grasslands. Further, 428 km² of croplands converted to forests during this period. In addition, large areas of croplands became urban land within/around Hohhot, Baotou, and Ordos after 2000 (Figures 4 and 5). These conversions led to a net decrease in croplands in the west.

### 3.3. Temporal and Spatial Effects of Cropland Evolution on Soil Wind Erosion Changes

In the whole of Inner Mongolia, the soil wind erosion modulus was generally higher in the west and lower in the east (Figure 6). The area with a moderate or more severe intensity level of wind erosion (soil wind erosion modulus > 2500 t/(km²·a)) was $2.96 \times 10^5$ km², accounting for approximately one-quarter of Inner Mongolia's area. In the three sub-regions, the average wind erosion modulus in the eastern, central and western parts was 402.71 t/(km²·a), 1665.92 t/(km²·a), and 7436.51 t/(km²·a), respectively. Further, in different land-cover types, the average wind erosion modulus was 1640.32 t/(km²·a), 1047.23 t/(km²·a), 321.92 t/(km²·a) and 8198.24 t/(km²·a) in croplands, grasslands, forests and bare lands, respectively.

From 1990 to 2018, the impact of the conversion of croplands to other land types (such as forests, grasslands and bare lands) on the soil wind erosion modulus caused a net reduction of 2.3 million tons. The impact of the conversion of grasslands/forests to croplands on soil wind erosion led to a reduction of 5.0 million tons, with 0.87 and 3.15 million tons from forests and grasslands, respectively. Although wind erosion events in the west were more severe compared to the eastern and central regions, approximately 80% of the increases on soil wind erosion still occurred in the east.

Meanwhile, the impact of the conversion of croplands to grasslands/ forests, or from bare lands to croplands, was the main reason for the decrease of soil wind erosion, with 62% of total soil wind erosion loss recorded in the eastern region and 25% in the western region (Figure 6). Conversion from bare lands to croplands led to an erosion modulus decrease of 0.71 million tons in the east and 0.20 million tons in the west. Finally, conversion from croplands to grasslands led to an erosion modulus decrease of 0.66 million tons in the east and 0.37 million tons in the west.

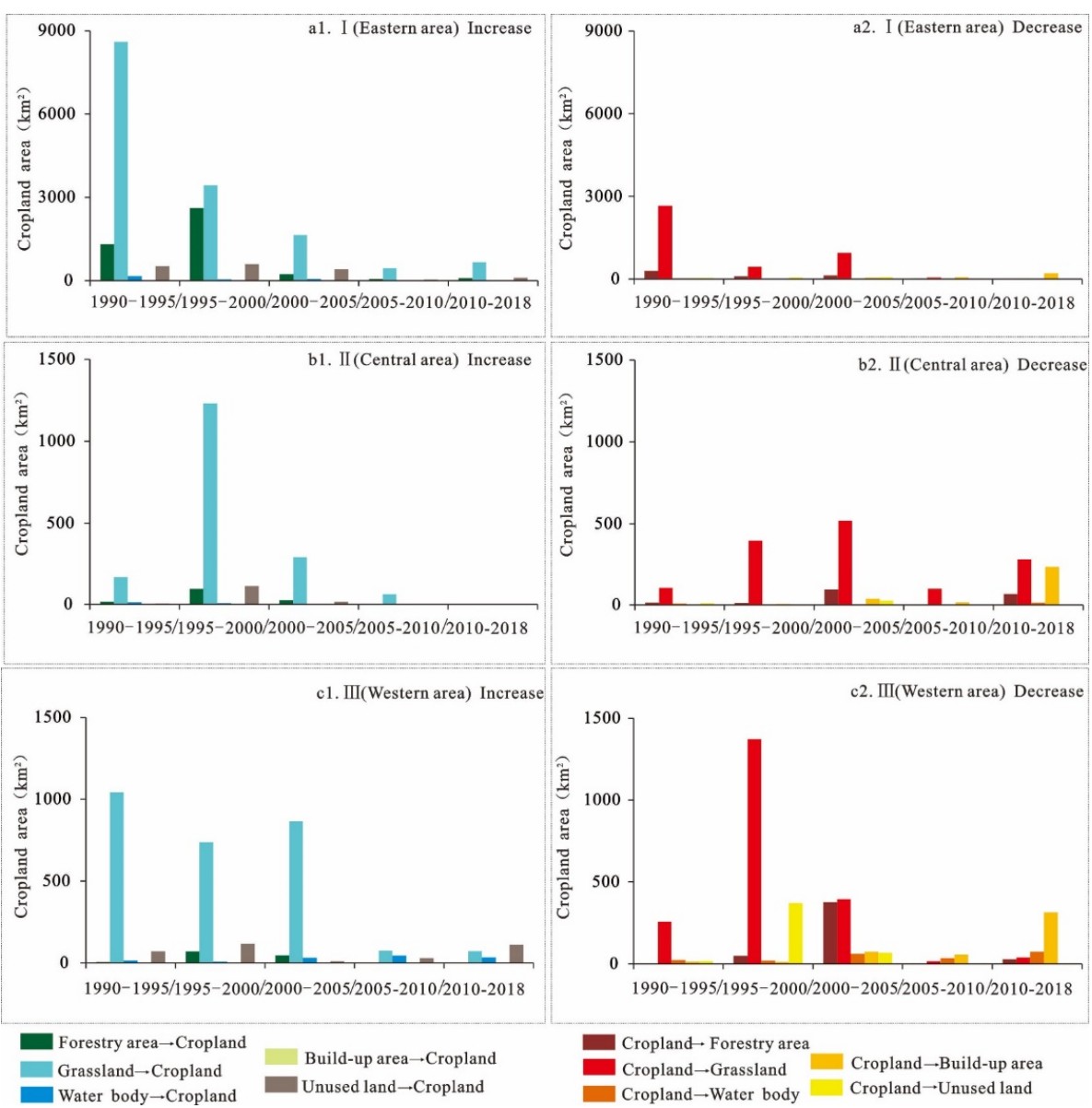

**Figure 4.** Conversions from or to croplands in three sub-regions of Inner Mongolia of China from 1990 to 2018.

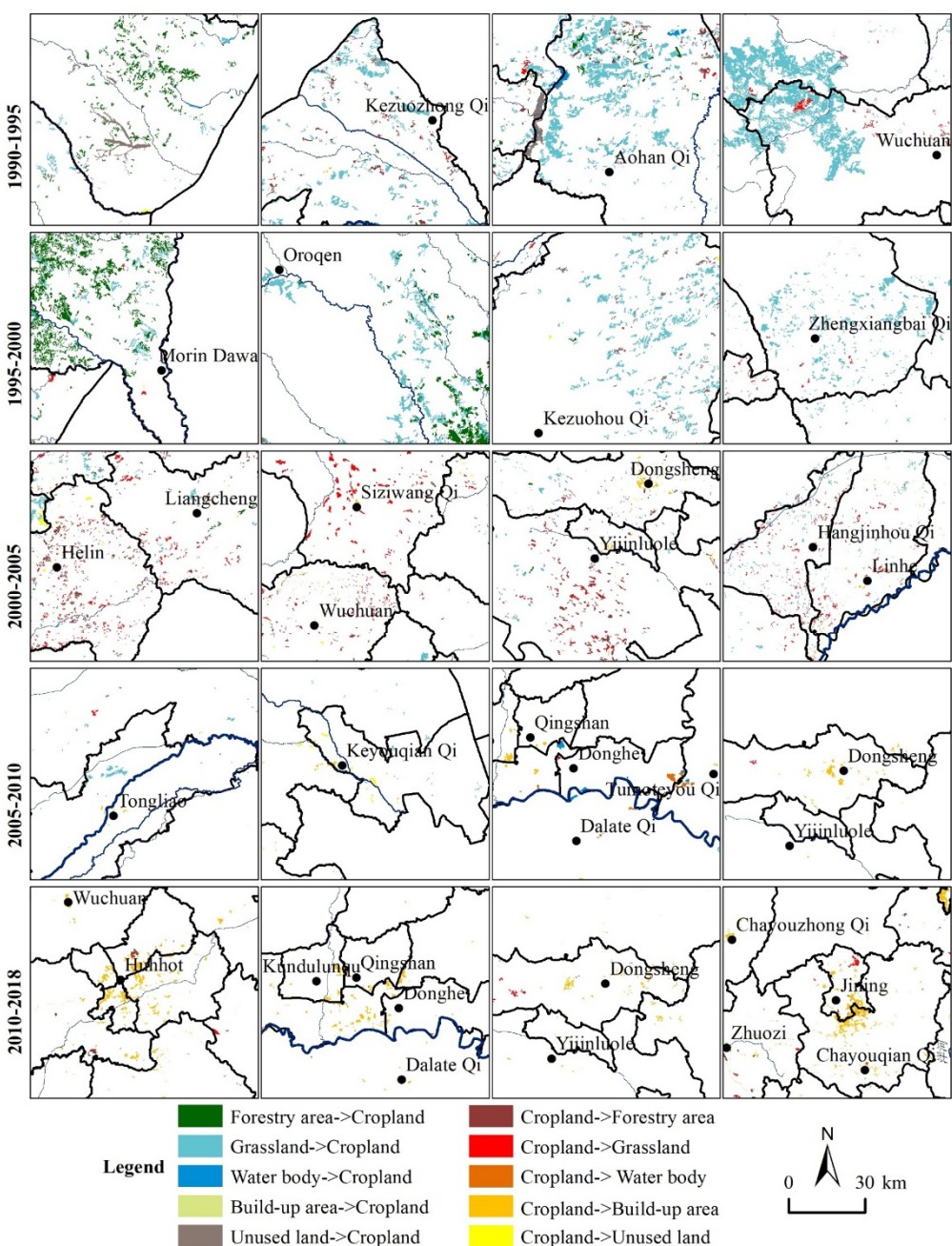

**Figure 5.** Spatial change patterns of the converted cropland types in typical areas of Inner Mongolia from 1990–2018. Please note: the symbol "->" represents the conversion direction between the two land types.

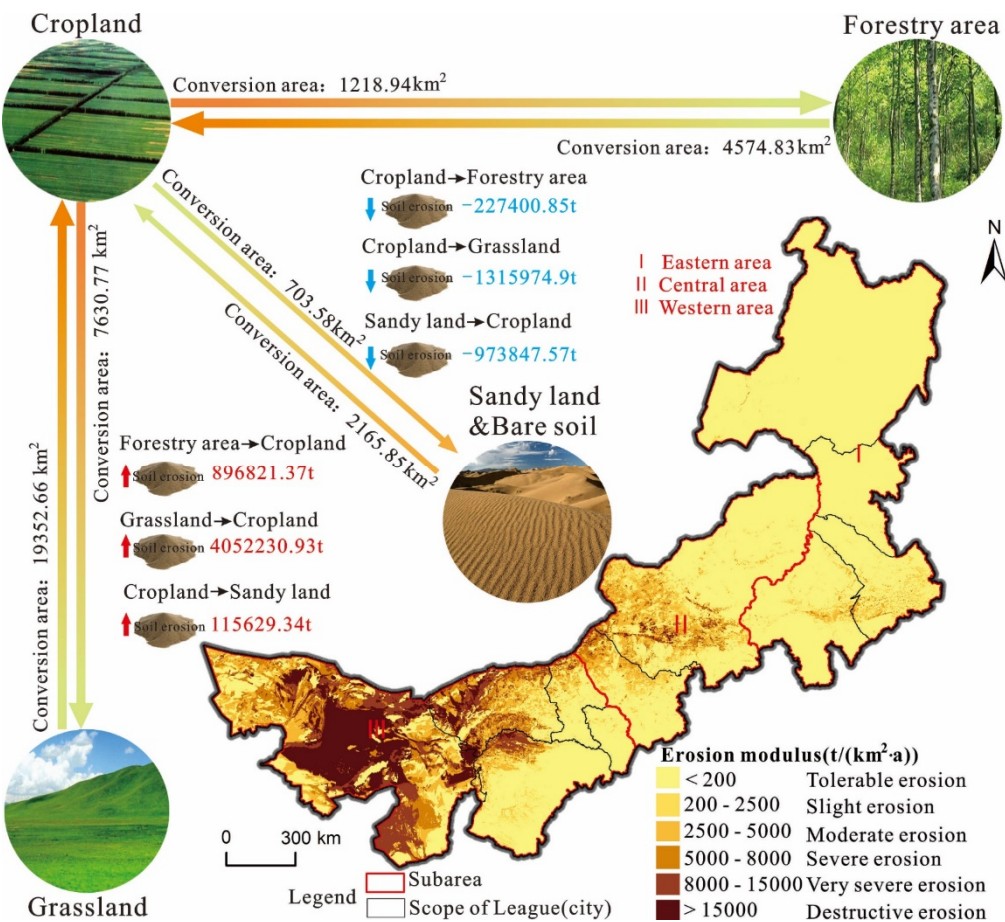

**Figure 6.** Effect of cropland change on soil wind erosion in Inner Mongolia from 1990–2018.

## 4. Discussions

### 4.1. The Wind Erosion Modulus and the Water Constraint for Sustainable Agricultural Development

This study investigated the impact of cropland evolution on soil wind erosion in Inner Mongolia of China. There was a net increase of 15,543 km$^2$ in cropland during 1990–2018, which was about 1.3% of the total area of Inner Mongolia (1.18 million km$^2$). Although the net increase of cropland area was relatively small, the conversion area of croplands and other land types happened in most regions of the study area (Figure 4). This means that soil erosion changes from land transformations between cropland and other land types had a wide influence on the study area. The average rate of soil erosion changes in each was higher in the study area than the global level [35]. Inner Mongolia is an ecological barrier area in North China, and soil wind erosion changes are important for agriculture and ecological environments. The soil wind erosion modulus from croplands in Inner Mongolia was approximately 646.5–2957.1 t/(km$^2$·a), with the low values in the east and the high values in the west from 1990 to 2018 (Figure 6). Our results are generally consistent with related researches conducted in dryland regions. Simulation in arid/semi-arid China found that the wind erosion modulus in croplands was approximately 375–2106 t/(km$^2$·a) [22]. Zhang [15] conducted field investigations in the southeastern border of Inner Mongolia and found that the wind erosion modulus in croplands was 109–4534 t/(km$^2$·a). Simulation in Central Asia indicated that the value was 421–527 t/(km$^2$·a), and will achieve 672–709 t/(km$^2$·a) in 2050 [4]. Compared with that, the wind erosion values from croplands in some sub-humid and humid regions were much lower or even zero due to abundant rainfall [22].

Though the wind erosion modulus in croplands was much lower than that in deserts, it was still higher than that in grasslands and forests [36,37]. Wind erosion from croplands was affected by climate, soil properties, surface characteristics, crop type and farming operations [36]. The former three factors were, to a great extent, determined by natural conditions, while the latter two were mainly determined by human operations. Farming operations, especially land preparation (i.e., plowing, leveling beds, planting, weeding), easily produced dust because it was conducted in spring when wind speed was high. Measures such as ridge tillage, crop rotation and keeping sufficient crop residues left in the fallow period have been tested and found to be effective practices for reducing potential wind erosion on croplands [13,38,39], and these measures should be popularized. No-till has also been found to be another way to reduce soil loss caused by water erosion and deflation [40–42].

The type of crop affects harvesting time and water consumption, which are both closely related to erosion. The proportion of irrigated croplands in Inner Mongolia has increased rapidly, ranging from 40% to 60% since the 1970s, with the water sources mainly being groundwater and a small amount of surface water [43]. In the past 30 years, the planting area of water intensive crops, such as corn, has also increased rapidly (Figure 7); and corn-based food production accounted for over 75% of total agricultural water consumption. Irrigation for potato, corn and other crops mainly relied on groundwater, but the amount of agricultural groundwater consumption went over the regional groundwater carrying capacity, with the overuse rate being more than 110% [43]. Although the issue of ecological security caused by large-scale agricultural irrigation has been recognized and investigated by researchers and relevant departments [38,39,43], the amount of agricultural irrigation has not been effectively controlled. For the goal of globally sustainable development by 2030, agricultural development in Inner Mongolia should be transformed from a "water for food at any cost" approach to a balance of water-food approach, to ensure ecological security.

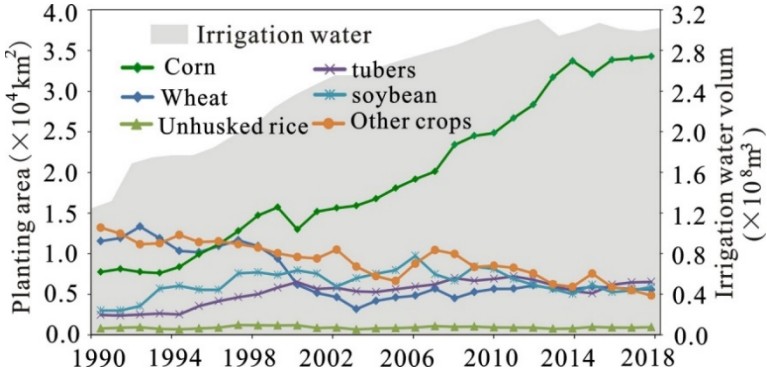

**Figure 7.** The changes of crop planting structure and irrigation water area in Inner Mongolia from 1990 to 2018.

### 4.2. The Effect of the Grain for Green Project on Eliminating Soil Wind Erosion

Studies have found that the soil wind erosion modulus showed a linear or exponentially decreasing law with an increase of vegetation coverage, canopy height, above-ground biomass, and species richness [22,44]. High coverage vegetation (e.g., shrub) had a strong effect on soil, effectively blocking air flow [45]. In Inner Mongolia, bare lands in spring and winter easily led to a high risk of soil erosion compared with woodlands and grasslands. The loss rate of organic carbon, total nitrogen, and total phosphorus during the soil erosion events was 18–38% in other regions [9,46]; however, we found that the soil wind erosion modulus increased by 5–15 times on average because of grasslands' conversion to croplands, especially in the central and western parts of Inner Mongolia. Land conversion from croplands to natural vegetation driven by the Grain for Green project effectively decreased soil wind erosion (Figure 6). This project also played an essential role in improving ecosys-

tem services such as erosion control and water retention [47,48]. However, decreases in soil erosion after 2000 still could not compensate for soil erosion increases because of the large area of grassland reclamation in the early 1990s. The fragile ecosystems in Inner Mongolia, especially in the west, make it challenging to find a balance between the development of croplands and the improvement of ecosystem services, such as erosion control, to achieve the sustainable management of agriculture, which should be our next research direction.

### 4.3. Cropland Evolution for Sustainable Agriculture and Food Security in Inner Mongolia

Cropland resources are essential for agricultural development. Increasing the production and utilization efficiency of croplands has become the primary approach to protect croplands and food security. In the past 30 years, cropland reclamation in Inner Mongolia has led to greater irrigation water demand and soil wind erosion, which has offset the effectiveness of the Grain for Green project since the 2000s. Large-scale cropland reclamation has led to severe regional land degradation and has exacerbated ecological vulnerability. Under the constraints of regional water resource shortages and over-exploitation of groundwater, the current trajectory of unfettered cropland reclamation is not in line with the goal of agricultural green development. This unsustainable trend would exacerbate the vulnerability of agriculture and threaten regional ecological security [2,43]. Therefore, it is important to explore the sustainable development path of agriculture in dryland regions (Figure 8). The central region of Inner Mongolia is dominated by grassland ecosystems, including the Xilingol steppe, Hunshandake Sandy Land, and Ulanqab desert steppe. The croplands here are fragile and of low quality, with water shortages and a high frequency of natural disasters such as drought being common. The key for managing croplands in the central region is to develop agriculture against a background of ecosystem protection. Specific paths towards this goal include projects such as water saving agriculture and the production of high-quality agriculture commodities based on regional water resource patterns. Moreover, the revegetation of croplands to forests/grasslands should be encouraged to achieve the comprehensive management of the ecological environment based on scientific evaluation.

In the western region of Inner Mongolia, croplands are mainly distributed on the Tumochuan Plain and the Hetao Plain. The main constraints for agricultural production here are water shortages, urban expansion, and land degradation, such as salinization and aeolian desertification. Here, the key for sustainable agriculture is to prioritize ecological protection. Croplands in some fragile zones should be returned to grasslands/shrubs. Agricultural development should consider intensive and high-efficiency agricultural facilities, along with an ecological-agricultural system that focuses on seedling cultivation.

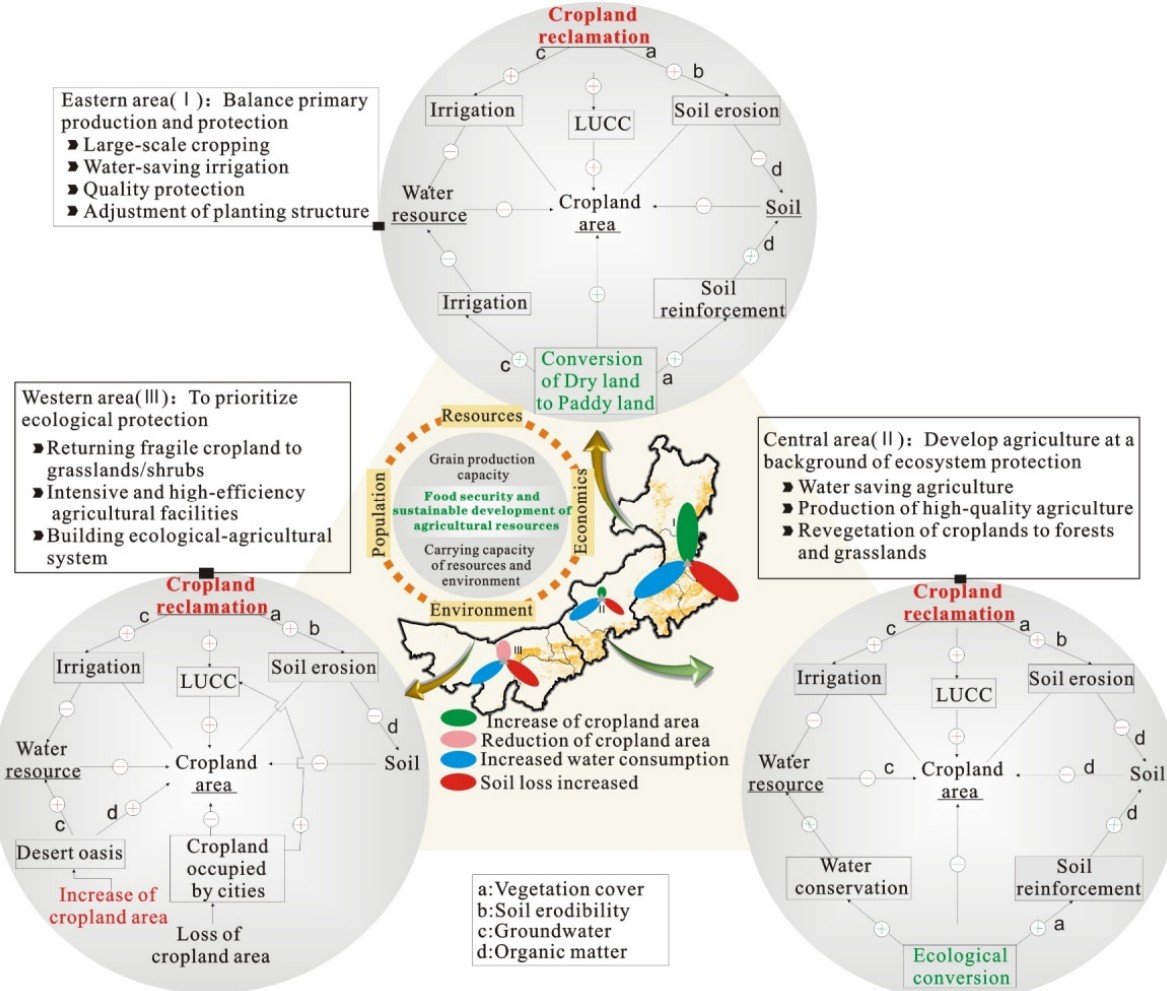

**Figure 8.** The paths for sustainable development of agriculture in Inner Mongolia.

## 4.4. Limitation of This Study and Future Directions

There were a few limitations in this study that should be addressed in the future. First, the modeling results obtained from the RWEQ model still contained considerable uncertainties, although we localized the parameters and did the validation based on field experiments. The retrieved value of soil wind erosion was significantly correlated with the field survey data, so it is effective for spatio-temporal pattern analysis and for analyzing the relative impacts of cropland changes. However, the accurate estimation of the wind erosion modulus for an area over a long period is still a challenge current studies face. Long term observations, high resolution data, and suitable up-scaling methods are needed to help construct a more robust model. Second, we selected Inner Mongolia, China as our case study to examine the impact of cropland evolution on wind erosion and also to propose a planning framework for sustainable agriculture. Based on sufficient data, a larger scale study on dryland regions, such as on the country or global scale are needed for a deeper understanding of the relevant scientific questions. However, Inner Mongolia, China, crossing 2400 km from the east to the west and covering arid, semi-arid and semi-humid regions, is very typical for dryland study. Our findings can be helpful for more ecologically and economically efficient policies aimed at sustainable cropland development and for improving the environments of dryland regions and beyond.

## 5. Conclusions

Cropland evolution, as well as its impact on soil wind erosion in the ecological barrier area of North China, Inner Mongolia, was detected in the years of 1990–2018, using the human-computer interaction method and the RWEQ model. We found that croplands showed a net increase of 15,542.92 km$^2$, with elements of spatial heterogeneities and temporal variations found. Croplands in eastern Mongolia showed continuous growth, but those in the west showed a decrease from 1995. The soil wind erosion modulus varied with the cropland evolution. Although the Grain for Green project led to a decline in soil wind erosion after 2000, the large-scale reclamation of grasslands/forests for planting crops still induced a net increase of the soil wind erosion modulus from 1990 to 2018. With the constraints of water shortages and the over-exploitation of groundwater in mind, we suggest a path based on the balance between "resource-production-ecosystem" to achieve sustainable and green agriculture development in these dryland regions. Further, the cropland space control policy should be strengthened by legislation to reduce wind erosion.

**Author Contributions:** W.C.: Conceptualization, methodology, software, writing original draft preparation. Y.Z.: Con-ceptualization, Writing-reviewing and editing. W.K.: Review and editing. T.P. and T.B.: Visualiza-tion and editing. J.Z. and L.J.: Editing. S.W.: Conceptualization and writing original draft prepara-tion. All authors have read and agreed to the published version of the manuscript.

**Funding:** This research was supported by the Science & Technology Basic Resources Investigation Program of China (2017FY101304), the National Key Research and Development Program of China (2016YFC0500204), the National Natural Science Foundation of China (4971130 & 42061069) and the Key technology and application of ecological quality diagnosis and integrated management of "Beautiful Inner Mongolia" (2019GG010). Cropland quality evaluation and cropland resources big data platform construction in Inner Mongolia. Research Foundation of Education Bureau of Inner Mongolia, China (NJYT-19-B29).

**Data Availability Statement:** Not applicable.

**Conflicts of Interest:** The authors declare no conflict of interest.

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
