# Peer review of "Impact of Cropland Evolution on Soil Wind Erosion in Inner Mongolia of China"

_land, doi:10.3390/land10060583_

Round 1

Reviewer 1 Report

The authors tackle an important issue “wind soil erosion” and its effects on resource management. The manuscript is well-written; however, it doesn’t discuss much about the impact of cropland evolution on wind soil erosion. It rather presents many other issues with land management in Inner Mongolia region. I have the following specific questions/comments:

Line 98-99: There is a typo: it should be 0.49 and 0.26 million km2

Line 169: How many field observations and measurements were conducted for such a vast area to justify the validation process? What parameters were measured? How were sampling locations selected? It seems that the field data for model validation was an important aspect of this work, but almost no information is provided about the field work.     

Line 179: It's not clear how rainfall erosion is separated from total erosion to get the net wind erosion. Needs clarification.

Line 251-254: Wind erosion has increase in both directions, i.e. when the land was converted from croplands to other land types and when it was converted from grasslands/forests to croplands? How could this be explained? If that’s the case what significant role cropland evolution plays in the region with regards to soil erosion?

Line 268: The Discussion section doesn’t specifically discuss the results of the current work. It rather discusses many other land management issues in the region. Many of the topics in this section are completely new and not related to the objective of the current work. There is no balance between the number of pages and amount of material presented in this section with the rest of manuscript. The Result section, for instance, is very short compare to Discussion section.

Line 383: There is a net increase of 15,543 km2 in cropland during 1990-2018. This is about 1.3% of the total area (1.18 million km2). So, how does this small change in cropland play a role in soil erosion in the region?

Author Response

Dear reviewer:

    Please see the attachment "Response to Reviewer1.pdf", thanks.

Reviewer 2 Report

Title: Impacts of cropland evolution on soil wind erosion in Inner Mongolia of China

Abstract: 
Line 24 - 25: "We selected Inner Mongolia of China, a typical water-limited but cropland expansion region" Could you please rephrase this statement to make it well understood?
Line 27 - 28: "Results showed that the cropland area of Inner Mongolia presented an increased trend from 1990-2018" Please which trend?
Line 28 - 30: "The characteristics of cropland changes in the eastern, central, and western parts of the region were continuous growth, balanced requisition-compensation, and decline, respectively." Could you please rephrase the statement and make it more clearer?
I suggest the abstract is revised to capture the study and made more clearer.

Introduction:
I suggest some comments on soil erosion's effect on crop production is provided

Materials and methods
Study area is well described
Could you please reformat the all equations and their numbering to be uniform?
Please check the format of Table 1

Discussion:
The section looks good but was mainly centered on the present study. Could you please fix in other similar studies to make the discussion more credible?

I find the present manuscript interesting and will appeal to readers. It presents trends of planting that can help to curb loss of soil (also loss of fertility). However, large portions of the texts are unclear or badly phrased and need revision. Also, the discussion section should include similar studies to present a solid case.

I therefore suggest the manuscript is thoroughly revised incorporating the comments raised to put it in a better state for publication.     

Author Response

Dear reviewer:

    Please see the attachment "Response to Reviewer2.pdf", thanks.

Author Response

Dear reviewer:

    Please see the attachment "Response to Reviewer3.pdf", thanks.

Round 2

Reviewer 2 Report

I appreciate the revision done by the authors.

Most of the concerns I raised have duly been addressed.

Aside minor grammar corrections, I find the current state of the manuscript acceptable for publication.